# Increased Neutrophil Percentage and Neutrophil–T Cell Ratio Precedes Clinical Onset of Experimental Cerebral Malaria

**DOI:** 10.3390/ijms241411332

**Published:** 2023-07-12

**Authors:** Lucas Freire-Antunes, Uyla Ornellas-Garcia, Marcos Vinicius Rangel-Ferreira, Mônica Lucas Ribeiro-Almeida, Carina Heusner Gonçalves de Sousa, Leonardo José de Moura Carvalho, Cláudio Tadeu Daniel-Ribeiro, Flávia Lima Ribeiro-Gomes

**Affiliations:** Laboratório de Pesquisa em Malária, Instituto Oswaldo Cruz & Centro de Pesquisa, Diagnóstico e Treinamento em Malária (CPD-Mal) of Fundação Oswaldo Cruz (Fiocruz) and of Secretaria de Vigilância em Saúde (SVS), Ministério da Saúde, Rio de Janeiro 21041-250, Brazil; lucasfreire.antunes@hotmail.com (L.F.-A.); uylagarcia@gmail.com (U.O.-G.); marcosrangel.bio@gmail.com (M.V.R.-F.); monicalucas.r@gmail.com (M.L.R.-A.); carinaheusner@hotmail.com (C.H.G.d.S.); leojmc@ioc.fiocruz.br (L.J.d.M.C.); malaria@fiocuz.br (C.T.D.-R.)

**Keywords:** malaria, cerebral malaria, neutrophils, T cells, neutrophil–lymphocyte ratio, NLR, *Plasmodium*, murine model, *Plasmodium berghei* ANKA

## Abstract

Newly emerging data suggest that several neutrophil defense mechanisms may play a role in both aggravating and protecting against malaria. These exciting findings suggest that the balance of these cells in the host body may have an impact on the pathogenesis of malaria. To fully understand the role of neutrophils in severe forms of malaria, such as cerebral malaria (CM), it is critical to gain a comprehensive understanding of their behavior and functions. This study investigated the dynamics of neutrophil and T cell responses in C57BL/6 and BALB/c mice infected with *Plasmodium berghei* ANKA, murine models of experimental cerebral malaria (ECM) and non-cerebral experimental malaria, respectively. The results demonstrated an increase in neutrophil percentage and neutrophil–T cell ratios in the spleen and blood before the development of clinical signs of ECM, which is a phenomenon not observed in the non-susceptible model of cerebral malaria. Furthermore, despite the development of distinct forms of malaria in the two strains of infected animals, parasitemia levels showed equivalent increases throughout the infection period evaluated. These findings suggest that the neutrophil percentage and neutrophil–T cell ratios may be valuable predictive tools for assessing the dynamics and composition of immune responses involved in the determinism of ECM development, thus contributing to the advancing of our understanding of its pathogenesis.

## 1. Introduction

Malaria is a parasitic disease that affects millions of people worldwide, especially in tropical and sub-tropical countries [1]. In 2021, over 247 million people were infected and 619,000 died from malaria. African children under the age of 5 are the most vulnerable group, accounting for 76% of all malaria deaths [1]. The disease is caused by protozoan parasites of the genus *Plasmodium*, with *P. falciparum* being the deadliest species [2]. The main complications of malaria include severe anemia, acute renal failure, liver damage, respiratory distress and cerebral malaria (CM) [3,4,5]. CM is a severe complication of *P. falciparum* infection characterized as an encephalopathy, which can cause seizures, impaired consciousness, coma, and death. Even with proper treatment, 15–20% of CM cases may progress to death [6,7], and those who survive may have severe neurocognitive sequelae and behavioral problems after the disease [8,9].

Despite extensive research, the exact mechanisms underlying the pathogenesis of CM remain poorly understood. However, it is known that the main pathophysiological events involve the adhesion of infected red blood cells (iRBCs) to endothelial cells in the cerebral microvasculature and the release of pro-inflammatory mediators into the circulation by immune cells activated by pathogen- and damage-associated molecular patterns (PAMPS and DAMPs). These events together amplify the activation and expression of adhesion molecules by endothelial cells, resulting in increased red blood cells (RBCs) and leukocytes sequestration on blood vessel walls, obstruction of blood flow, activation of intravascular coagulation, disruption of blood–brain barrier integrity, and ultimately neuroinflammation [10]. In other words, although the immune response is essential for parasite control and host protection, an excessive pro-inflammatory immune response may contribute to the pathogenesis of CM in some individuals [11].

Great progress has been made in understanding the adaptive immune response elicited during *Plasmodium* spp. infection. CD8 T cells, important cells of the adaptive immunity, have been implicated in the pathogenesis of CM [12]. These cells are recruited to the brain and contribute to the breakdown of the blood–brain barrier, which is a hallmark of CM [13]. However, it is quite intriguing that the role of innate immunity has received less attention, particularly the neutrophils. These cells are widely present in the human bloodstream and play a key role in the host’s first line of protection from microbial infections. In CM patients, there is evidence of neutrophil activation and granule protein release as well as the formation of neutrophil extracellular traps (NETs) [14,15,16]. However, it is yet to be determined whether these processes represent a suitable response to infection or an inadequate response that contributes to inflammation, tissue damage, and RBC sequestration. Moreover, the studies that have explored the role of neutrophils in the development of experimental CM (ECM) present contradictory findings [17,18].

In malaria, neutrophilia and lymphopenia are common immunological changes that indicate immune system disruption. Neutrophils containing hemozoin, a malaria pigment, have been associated with the development of CM and an increased risk of mortality in pediatric patients with severe malaria [19]. The neutrophil–lymphocyte ratio (NLR) value has also been investigated as a prognostic marker for clinical outcomes in malaria [20,21]. Studies have found a positive correlation between NLR and parasitemia [20,21] as well as disease severity [21].

The use of experimental models has been instrumental in advancing our understanding of the pathophysiology of CM. This is because experimental models allow us to understand the disease in a controlled environment, enabling us to investigate the role of specific cells and immune pathways in disease progression in different organs and tissues of the body. Although not reproducing all aspects of human disease, C57BL/6 mice infected with *P. berghei* ANKA, an experimental murine model of CM (ECM), share numerous immunological and histological features observed in CM patients [5,22] and develop clinical signs of neurological complications and cognitive–behavior sequelae [23,24,25].

In this study, we employed murine models of ECM and non-cerebral malaria by infecting C57BL/6 and BALB/c mice with *P. berghei* ANKA, respectively, to assess the dynamics of neutrophil and T cell responses as well as their ratios (neutrophil–T cell ratios), in the blood and spleen from the early stages of infection until death by ECM of the susceptible model.

## 2. Results

### 2.1. The Development of Experimental Cerebral Malaria Is Unrelated to Peripheral Parasitemia

C57BL/6 and BALB/c mice are well-known experimental models used to study ECM and non-ECM, respectively, upon infection with *P. berghei* ANKA [26]. C57BL/6 mice develop ECM and typically die 6–9 days after *P. berghei* ANKA infection [27], while most parasitized BALB/c mice develop anemia and eventually succumb to infection [28].

We examined the progression of body temperature and parasitemia in BALB/c and C57BL/6 mice following injection of iRBCs. By day 6 post-infection, C57BL/6 mice showed a sharp drop in body temperature, with most mice exhibiting severe hypothermia (less than 36 °C) (Figure 1A). In contrast, BALB/c mice only had a slight decrease in temperature. Despite the significant difference in body temperature between infected BALB/c and C57BL/6 mice on day 6 post-infection (Figure 1A), parasitemia increased similarly in both animal strains throughout the analyzed days of infection (Figure 1B). These data suggest that parasite replication per se is not associated with the development of ECM.

### 2.2. The Percentage of Neutrophils in the Spleen Significantly Increases Prior to the Development of Clinical Signs of Experimental Cerebral Malaria

The percentages of neutrophils and T cells in the spleen were also evaluated over the course of infection. These cells were identified based on their expression of specific surface markers as follow: neutrophils (CD11b^+^Ly6G^+^); T cells (CD3^+^); CD4 T cells (CD3^+^CD8^−^CD4^+^); and CD8 T cells (CD3^+^CD4^−^CD8^+^) (Figure 2A). In parasitized BALB/c mice, the percentage of neutrophils in the spleen decreased over time, whereas in C57BL/6 mice, the percentage of neutrophils increased significantly on day 4, which was followed by a decrease on day 6 post-infection (Figure 2B). On the other hand, the percentage of T cells, which comprised approximately 31–37% of splenocytes in naïve mice of both strains, decreased to around 22–23% on day 4 and 6 post-infection (Figure 2C). Analysis of CD4 and CD8 T cell populations also revealed a reduction in their percentage values after infection in both BALB/c and C57BL/6 mice (Figure 2D,E).

Given the significant increase in the percentage of neutrophils in C57BL/6 mice on day 4 post-infection and previous studies suggesting the use of NLR as a predictor of malaria severity [21,29], we investigated the values of the neutrophil–T cell ratio (Figure 3A), neutrophil–CD4 T cell ratio (Figure 3B), and neutrophil–CD8 T cell ratio (Figure 3C) in the spleens of non-infected and infected mice. On day 4 post-infection, C57BL/6 mice, the classic ECM model, showed an increase in all three neutrophil–T cell ratios compared to their respective naïve mice and BALB/c mice at the same time point of infection. However, no difference in neutrophil–T cell ratios was observed in the different assessments during BALB/c mice infection (Figure 3A–C).

### 2.3. Increased Neutrophils Percentage and Neutrophil–T Cell Ratios in the Blood of Parasitized C57BL/6 Mice Precede the Development of ECM

To investigate whether the increase in the neutrophil percentage and the neutrophil–T cell ratios observed in the spleen of C57BL/6 mice on day 4 post-infection was also present in the peripheral blood, we assessed the percentages of neutrophil, T cells, and their subpopulations (CD4 T cells and CD8 T cells) in the blood of BALB/c and C57BL/6 mice. On day 4 post-infection, there was an increase in the percentage of neutrophil in C57BL/6 mice compared to naive mice, while the percentages of total T cells and CD8 T cells decreased (Figure 4A–D). In infected BALB/c mice, only the percentage of CD8 T cells decreased (Figure 4D). Blood analysis also revealed an increase in the neutrophil–T cell ratio, neutrophil–CD4 T cell ratio and neutrophil–CD8 T cell ratio in the ECM-susceptible model. No statistical difference was observed between parasitized and naïve BALB/c mice in the different analyses of neutrophil–T cell ratio (Figure 5A–C).

## 3. Discussion

Malaria is a treatable and curable disease. However, despite this, hundreds of thousands of people still die or are left with neurological and cognitive sequelae after the severe forms of the disease. Therefore, understanding the cells and events involved in the pathophysiology of the disease is mandatory [30]. In this study, we took advantage of well-established models of ECM and non-cerebral experimental malaria to investigate the initial events of the neutrophil and T cell response, early in infection. Our data demonstrate that (1) clinical signs of ECM, such as hypothermia and paralysis, occur suddenly in *P. berghei* ANKA-infected C57BL/6 animals and are rapidly followed by death of the animals, despite no observable difference in peripheral parasitemia between these animals and the non-susceptible BALB/c mouse model; (2) the development of ECM is preceded by an increase in the neutrophil percentage and neutrophil–T cell ratios in the spleen and blood of parasitized mice.

In the scientific literature, there is a lack of consensus regarding the levels of parasitemia observed between susceptible and resistant models of ECM. Several studies have reported similar parasitemia levels between C57BL/6 and BALB/c mice infected with *P. berghei* ANKA; others describe higher levels in BALB/c when compared to C57BL/6 mice and vice versa [26,31,32,33,34,35]. These conflicting findings between studies may be associated with the analysis of peripheral parasitemia, as the retention of iRBCs in the capillaries of different organs can compromise the accuracy of parasitemia [36]. Within our analysis on experimental malaria, no association was found between peripheral parasitemia and the development of CM and, on the contrary, the two murine experimental models showed the same parasitemia levels when evaluated on different days of infection followed until the death of the infected C57BL/6 mice.

Hematological parameters are frequently altered in infectious diseases, including malaria. Most malaria patients show lymphopenia and increased neutrophil counts compared to non-infected individuals [20,29]. In our study, we also analyzed the percentage of T cells and neutrophil populations in ECM and non-cerebral experimental malaria models and observed early alterations in the percentage of these cells in the spleen and blood following infection. Both experimental models showed a reduction in the percentage of T cells in the spleen, while an increase in the percentage of neutrophils was found only in C57BL/6 mice at day 4 post-infection. Blood analysis also revealed an early increase in neutrophil percentage and a reduction in T cells in infected C57BL/6 mice at day 4 post-infection. BALB/c mice showed only a reduction in the percentage of CD8 T cells in the blood. These findings suggest that there are differences in hematological parameters between susceptible and non-susceptible animals to ECM, indicating that these parameters may serve as useful indicators of disease susceptibility and underlying pathogenetic mechanisms.

Elevated NLR is currently used to diagnose and/or predict the outcome of cancer, cardiovascular disorders, and infectious diseases [37,38,39,40], and its level also increases with age [41] and in chronically stressed mice [42]. Nonetheless, the applicability of NLR as a diagnostic and prognostic tool for patients with severe malaria remains disputed. A study found no discernible contrasts between severe and non-severe malaria patients [20], whereas others have demonstrated that individuals with severe malaria exhibit a greater NLR than healthy individuals [21,43]. Interestingly, in a controlled human experimental study of *P. falciparum* infection conducted with healthy non-immune volunteers, it was observed that the initial appearance of parasites in the peripheral blood is accompanied by a marked increase in NLR [44]. In the context of our murine experimental study, we found that C57BL/6 mice infected with *P. berghei* ANKA displayed an increase in neutrophil–T cell ratios in the spleen and blood before the development of ECM, while BALB/c mice showed no changes in these values during infection.

To the best of our knowledge, this study is the first to investigate the ratio of neutrophils to T cells and to observe an increase in this ratio before the onset of clinical signs in experimental cerebral malaria. Although we have not examined this ratio in other experimental models of cerebral malaria, such as CBA and Swiss mice infected with *P. berghei* ANKA [45,46,47,48], which could be a limitation of our study, we would like to point out that the experimental model used in this study is the most widely accepted and used model in the literature as it shares several characteristics with human cerebral malaria.

The hypothermia and neutrophil mobilization observed in our study are not unique to ECM; they are also observed in certain, but not all, pathologies that utilize the C57BL/6 mouse. This includes models such as of viral infections [49,50,51,52,53,54], acute sepsis [55,56], and studies evaluating the toxicity of staphylococcal enterotoxin [57,58].

In summary, our results suggest that the neutrophil percentage and neutrophil–T cell ratios could represent biomarkers that can foresee the onset of CM in the experimental malaria model. Additionally, our findings offer valuable insights into under-explored aspects of immune response, enhancing our understanding of disease pathogenesis and expanding knowledge about the mechanisms governing immune responses during malaria infection, thereby addressing significant gaps in the scientific literature.

## 4. Materials and Methods

### 4.1. Animals and Ethics Statement

Six to eight-week-old female C57BL/6 and BALB/c mice, weighing 16–20 g, were provided by the Fiocruz’s Institute of Science and Technology in Biomodels (ICTB-Fiocruz, Rio de Janeiro, Brazil). Mice were kept in ventilated cages with free access to food and water, on a 12/12 h light/dark cycle, and in a specific pathogen-free room at the *Instituto Oswaldo Cruz* (IOC-Fiocruz) with constant temperature. This study was carried out in strict accordance with guidelines and regulations, and the Fiocruz Animal Welfare Committee permitted the experiments (license number L-041/2016-A2).

### 4.2. Parasite and Infection

Infection was performed with *P. berghei* ANKA expressing a green fluorescent protein (GFP), which was a grant from the Malaria Research and Reference Reagent Resource Center (MR4 number: MRA-865). Mice were inoculated intraperitoneally (i.p) with 1 × 10^6^ iRBCs.

### 4.3. Clinical Parameters

On day 4 and 6 post-infection, parasitemia and rectal temperature were evaluated. Parasitemia was assessed by flow cytometry using a small blood sample collected through a prick on the tail end of the mouse, which was then diluted in phosphate-buffered saline (PBS). The percentage of iRBCs (RBCs expressing GFP) was calculated based on the analysis of 10,000 RBCs. Rectal temperature was measured using a thermocouple probe (Oakton^®^ Acorn^TM^; Oakton Instruments, Vernon Hills, IL, USA).

### 4.4. Tissue Processing and Immunophenotyping

To assess the percentages of neutrophils and T cells in the blood, 70 μL of blood was collected from the tail tip or by cardiac puncture from the animals. To evaluate splenic neutrophil and T cell percentages, after blood withdrawal and euthanasia, infected and naïve mice were subjected to cardiac perfusion with 20 mL of cold PBS. Spleens were removed and mechanically dissociated, and the resulting single-cell suspension from spleen and blood was treated with lysis buffer (LGC Biotechnology, Hoddesdon, UK) to remove RBCs. Blood leukocytes and approximately 1 × 10^6^ spleen cells were incubated with anti-Fc-γ III/II (CD16/32) receptor Ab (2.4G2, BD Biosciences, Franklin Lakes, NJ, USA) and a pool of fluorochrome-conjugated antibodies diluted in PBS containing 5% of fetal bovine serum (FBS). The following antibodies were used: Alexa Fluor 700 anti-mouse CD11b (M1/70, BD); APC anti-mouse Ly6G (1A8, BD); Percp-Cy5.5 anti-mouse CD3 (145-2C11, BD); APC-H7 anti-mouse CD4 (GK1.5, BD); PE-Cy7 anti-mouse CD8 (53–6.7, BD). Cells were incubated for 30 min at 4 °C, protected from light, according to the manufacturer’s instructions. After incubation, cells were washed to remove unbound antibodies. Samples were acquired using a CytoFLEX S flow cytometer (Beckman Coulter, Pasadena, CA, USA), and data analysis was performed using the FlowJo 10.0 program (BD Biosciences).

### 4.5. Statistical Analysis

All graphs and statistical analyses were performed using GraphPad Prism software version 8.0. Data are presented as the mean ± standard error of the mean. Comparisons between groups were made by two-way ANOVA with Tukey’s multiple comparisons test. When appropriated, the unpaired Student’s *t*-test with a 95% confidence level was used to compare the control and infected groups of the same mouse strain. Differences with a *p*-value less than 0.05 were considered statistically significant.

## Figures and Tables

**Figure 1 ijms-24-11332-f001:**
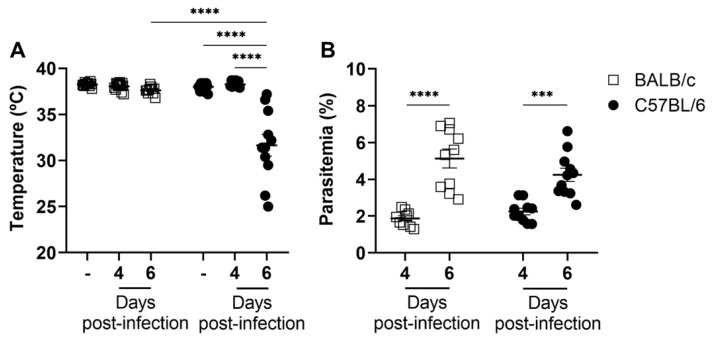
Changes in body temperature and parasitemia during *P. berghei* ANKA infection in BALB/c and C57BL/6 mice. Mice were infected with 1 × 10^6^ iRBCs. Body temperature (**A**) and parasitemia (**B**) were measured in non-infected and/or *P. berghei* ANKA-infected mice on days 4 and 6 post-infection. Following *P. berghei* ANKA infection, only the C57BL/6 mice exhibited severe hypothermia, although parasitemia levels were similar in both mouse strains. The non-infected group is represented by the symbol “-” on the *x*-axis. Data are from 2 independent pooled experiments with 11–16 animals/group; *p* = 0.0009 (***), *p* < 0.0001 (****).

**Figure 2 ijms-24-11332-f002:**
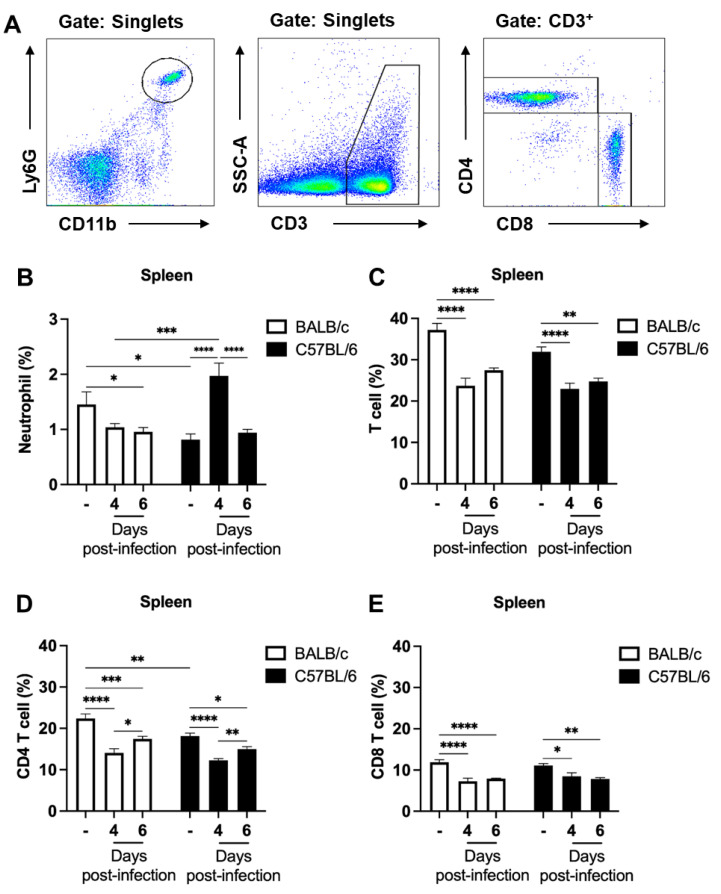
Kinetics of neutrophil and T cell mobilization following *P. berghei* ANKA infection in BALB/c and C57BL/6 mice. Mice were injected with 1 × 10^6^ iRBCs. (**A**) Representative dot plots used to identify the neutrophils (CD11b^+^Ly6G^+^), T cells (CD3^+^), CD4 T cells (CD3^+^CD8^−^CD4^+^) and CD8 T cells (CD3^+^CD4^−^CD8^+^) recovered from the spleen at different times of infection. Changes in neutrophils (**B**), total T cells (CD3^+^ cells) (**C**), CD4 T cells (**D**) and CD8 T cells (**E**) expressed as percentage of total spleen cells of non-infected and *P. berghei* ANKA-infected mice on days 4 and 6 post-infection. Infected BALB/c mice exhibited a decrease in the percentage of splenic neutrophils over time, while C57BL/6 mice showed a significant increase on day 4 post-infection. Additionally, the percentage of T cells decreased in both strains, including CD4 and CD8 T cell populations. The non-infected group is represented by the symbol “-” on the *x*-axis. Data are from two independent pooled experiments (10–11 animals per group); *p* < 0.05 (*), *p* < 0.005 (**), *p* < 0.0005 (***), *p* < 0.0001 (****).

**Figure 3 ijms-24-11332-f003:**
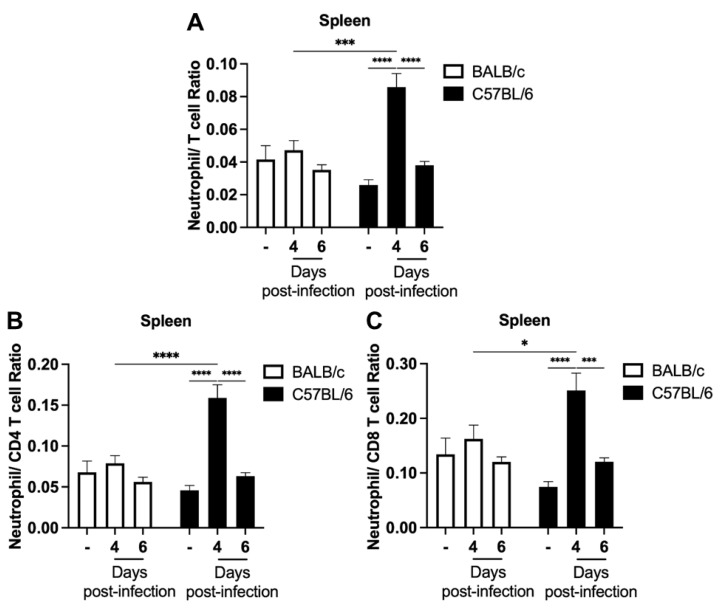
Kinetics of neutrophil–T cell ratio in the spleen following *P. berghei* ANKA infection in BALB/c and C57BL/6 mice. Mice were injected with 1 × 10^6^ iRBCs, and the neutrophil–T cell ratio (**A**), neutrophil–CD4 T cell ratio (**B**) and neutrophil–CD8 T cell ratio (**C**) were evaluated based on the percentage of neutrophils and T cells present in the spleen of non-infected and *P. berghei* ANKA-infected mice on days 4 and 6 post-infection. Notably, an early increase in the splenic neutrophil–T cell ratio was observed during *P. berghei* ANKA infection in C57BL/6 mice. The non-infected group is represented by the symbol “-” on the *x*-axis. Data are from two independent pooled experiments (10–11 animals/group); *p* < 0.05 (*), *p* < 0.0005 (***), *p* < 0.0001 (****).

**Figure 4 ijms-24-11332-f004:**
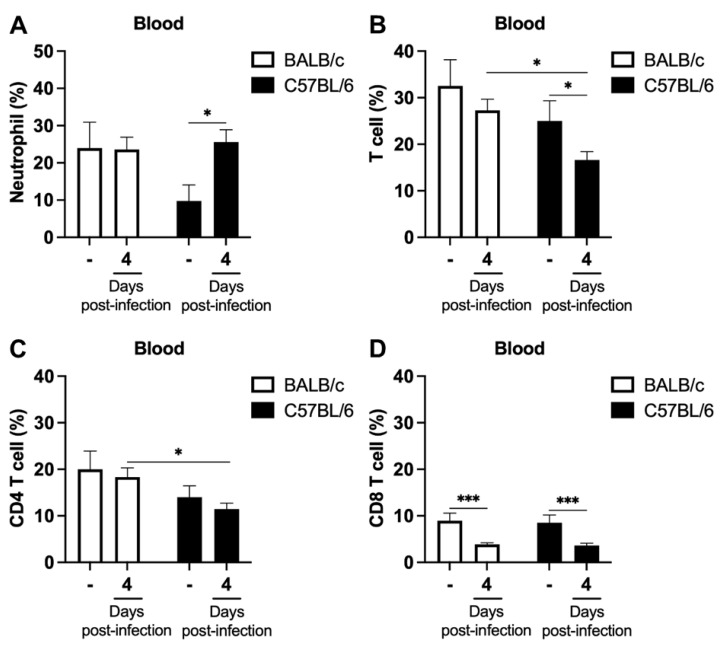
Mobilization of neutrophils and T cells in the blood of BALB/c and C57BL/6 mice following infection with *P. berghei* ANKA. Mice were injected with 1 × 10^6^ iRBCs, and the percentage of neutrophils (**A**), T cells (**B**), CD4 T cells (**C**) and CD8 T cells (**D**) in the blood of non-infected and *P. berghei* ANKA-infected mice on day 4 post-infection were analyzed. Most notably, an increase in neutrophils was observed in blood of C57BL/6 mice infected with *P. berghei* ANKA. The non-infected group is represented by the symbol “-” on the *x*-axis. Data are from 2 independent pooled experiments (total of 7 non-infected animals/group and 21 infected animals/group); *p* < 0.05 (*), *p* < 0.0005 (***).

**Figure 5 ijms-24-11332-f005:**
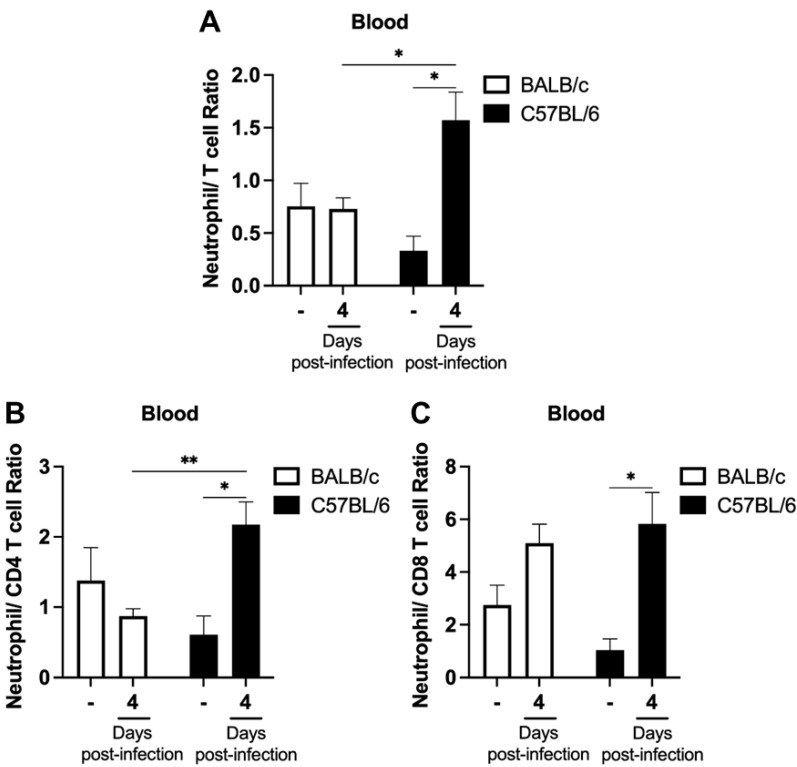
Kinetics of neutrophil–T cell ratio in the blood following *P. berghei* ANKA infection in BALB/c and C57BL/6 mice. Mice were injected with 1 × 10^6^ iRBCs, and the ratios of neutrophil–T cell ratio (**A**), neutrophil–CD4 T cell ratio (**B**) and neutrophil–CD8 T cell ratio (**C**) were assessed in the blood of non-infected and *P. berghei* ANKA-infected mice on day 4 post-infection. Notably, the neutrophil–T cell ratio in the blood showed an early increase in C57BL/6 mice infected with *P. berghei* ANKA. The non-infected group is represented by the symbol “-” on the *x*-axis. Data are from two independent pooled experiments (total of 7 non-infected animals/group and 21 infected animals/group). *p* < 0.05 (*), *p* < 0.005 (**).

## Data Availability

The article includes the data that support its conclusions. The raw dataset generated during the study can be obtained from the corresponding author upon reasonable request.

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
