# Peer review of "Increased Neutrophil Percentage and Neutrophil–T Cell Ratio Precedes Clinical Onset of Experimental Cerebral Malaria"

_ijms, 2023, doi:10.3390/ijms241411332_

Round 1

Reviewer 1 Report

The manuscript "Increased Neutrophil Percentage and Neutrophil-T Cell Ratio  Precedes Clinical Onset of Experimental Cerebral Malaria" by Antunes et. al, explores a significant underexplored research topic of the progression of cerebral malaria. The immune response in model murine models upon infection and the correlation of innate immune response with the outcome of infection. The manuscript is well written and data supports the conclusion presented by the authors. It is important to further the understanding of the development of cerebral malaria and the manuscript touches upon the significant role neutrophils might be playing in the process.

Author Response

Reviewer 1

“The manuscript "Increased Neutrophil Percentage and Neutrophil-T Cell Ratio  Precedes Clinical Onset of Experimental Cerebral Malaria" by Antunes et. al, explores a significant underexplored research topic of the progression of cerebral malaria. The immune response in model murine models upon infection and the correlation of innate immune response with the outcome of infection. The manuscript is well written and data supports the conclusion presented by the authors. It is important to further the understanding of the development of cerebral malaria and the manuscript touches upon the significant role neutrophils might be playing in the process”.

Thank you for your kind comments. We appreciate your recognition of the importance of this study. We are glad to hear that the manuscript is well written and that the data support the conclusions presented. We agree that further understanding of the development of cerebral malaria is critical, and it is encouraging to see the manuscript shed light on the possible role of neutrophils in this process.

The complete letter, containing the responses to all reviewers, and the revised version of our manuscript are attached.

Reviewer 2 Report

Report 1

The paper describes the dynamics of neutrophil and T cell responses in C57BL/6 and BALB/c mice infected with Plasmodium berghei ANKA, as well as their ratios (neutrophil: T cell ratios), in blood and spleen. Major results demonstrated an increase in neutrophil percentage and neutrophil T-cell ratios in the spleen and blood before the development of clinical signs of ECM (in C57Bl/6 mice).

Recently data have been suggested that several neutrophil defense mechanisms may play a dual role in malaria infection; 2) however, the studies that have explored the role of neutrophils in the development of ECM present contradictory findings and 3) the neutrophil-lymphocyte ratio (NLR) value has also been investigated as a prognostic marker for clinical outcomes in malaria.

So, this paper helps to fill the gap and provides more details on the successive clinical and hematological events that followed ECM until the death of the susceptible mice.  However, there are several things authors may need to correct and clarify in the paper. They are:

1. In the Abstract, line 18, the authors wrote: "The results demonstrated an increase in the percentage of neutrophils and the neutrophil-T-cell ratio in the spleen and blood before the development of clinical signs of ECM. Were these results not observed in the non-susceptible model of cerebral malaria?  If yes or no, add this information to the text on lines 18/19.

2. Review the nomenclature for scientific names. It is not necessary to indicate the abbreviation of the genus name, as the authors did in line 34. The second time the organism's name is written, the genus can be abbreviated. The species should never be abbreviated.

3. Regarding the legends of all figures, remove all information regarding the methods and statistical tests used, as these are provided in the Materials and Methods section. The legend texts should be improved. The legend should tell the reader what the graphs/images show.

4. Indicate in the legend of all figures that contain this information, that the non-infected group is represented by the dash (-) on the x-axis.

5. Review the titles of Figures 3, 4 and 5: they are inconsistent with the information in the legend regarding the two animal strains. They are more appropriate to be included as part of the text than in the title.

6. Indicate in line 159 in which figure is found the information regarding the Balb/c mice.

7.In line 225, change “experiment” to “experimental”.

8. The authors should inform/discuss in the manuscript if the results obtained are specific for cerebral malaria or if C57Bl/6 mice present all the alterations described against any pro-inflammatory condition. They should also inform/discuss if other experimental models of cerebral malaria, if any, also course with the observed alterations concerning decrease in body temperature, neutrophilia and increase in NLR. This information could appear as limitations of the study developed.

9. It might be interesting to perform all the experiments done with the two strains of mice by initiating malaria infection with sporozoites. According to reference 43, humans infected with sporozoites showed a marked increase in NLR as soon as parasites appear in the blood stream.

10. The description of the statistical analyses performed should be improved.

Author Response

Reviewer 2

The paper describes the dynamics of neutrophil and T cell responses in C57BL/6 and BALB/c mice infected with Plasmodium berghei ANKA, as well as their ratios (neutrophil: T cell ratios), in blood and spleen. Major results demonstrated an increase in neutrophil percentage and neutrophil T-cell ratios in the spleen and blood before the development of clinical signs of ECM (in C57Bl/6 mice).

Recently data have been suggested that several neutrophil defense mechanisms may play a dual role in malaria infection; 2) however, the studies that have explored the role of neutrophils in the development of ECM present contradictory findings and 3) the neutrophil-lymphocyte ratio (NLR) value has also been investigated as a prognostic marker for clinical outcomes in malaria.

So, this paper helps to fill the gap and provides more details on the successive clinical and hematological events that followed ECM until the death of the susceptible mice. 

We are grateful for your recognition of our study. Thank you for your comments and support.

However, there are several things authors may need to correct and clarify in the paper. They are:

  1. In the Abstract, line 18, the authors wrote: "The results demonstrated an increase in the percentage of neutrophils and the neutrophil-T-cell ratio in the spleen and blood before the development of clinical signs of ECM. Were these results not observed in the non-susceptible model of cerebral malaria? If yes or no, add this information to the text on lines 18/19.

We appreciate your valuable contribution to improving the clarity of our article. In the revised version of the Abstract, lines 18-20, we have included the information “a phenomenon not observed in the non-susceptible model of cerebral malaria”. This clarification helps differentiate the findings in the susceptible model from the findings in the non-susceptible model of cerebral malaria. Thank you for bringing this to our attention.

  1. Review the nomenclature for scientific names. It is not necessary to indicate the abbreviation of the genus name, as the authors did in line 34. The second time the organism's name is written, the genus can be abbreviated. The species should never be abbreviated.

We have carefully reviewed and revised the nomenclature in line 34 of the manuscript, following your suggestion. We removed the abbreviation of the genus name and ensured that the species name was not abbreviated. We are grateful for your contribution to improving accuracy and adherence to scientific standards in our work.

  1. Regarding the legends of all figures, remove all information regarding the methods and statistical tests used, as these are provided in the Materials and Methods section. The legend texts should be improved. The legend should tell the reader what the graphs/images show.

Based on your suggestion, we have removed all redundant information related to the methods and statistical tests used, as these details are already provided in the Materials and Methods section. We have also revised the legends of all figures to focus on providing a clear description of what the graphs represent.

We believe that these changes will improve the readability and overall understanding of the figures.

  1. Indicate in the legend of all figures that contain this information, that the non-infected group is represented by the dash (-) on the x-axis.

Based on your comments, we have updated the legends for all figures with the sentence “The non-infected group is represented by the symbol "-" on the x-axis”. This will provide clarity and help readers to interpret the figures accurately.

  1. Review the titles of Figures 3, 4 and 5: they are inconsistent with the information in the legend regarding the two animal strains. They are more appropriate to be included as part of the text than in the title.

After careful consideration, we agreed that information on the two animal strains would be most appropriate. We have made the necessary revisions to ensure consistency and clarity in the titles of Figures 3, 4, and 5.

  1. Indicate in line 159 in which figure is found the information regarding the Balb/c mice.

We have revised and clarified in the text that the information about the Balb/c mice can be found in Figure 4D, as indicated in line 167-168. This will allow readers to easily locate the relevant information in the figures.

7.In line 225, change “experiment” to “experimental”.

We have made the correction as suggested. In line 238, we have changed "experiment" to "experimental" to ensure accurate and proper use of the language.

  1. The authors should inform/discuss in the manuscript if the results obtained are specific for cerebral malaria or if C57Bl/6 mice present all the alterations described against any pro-inflammatory condition. They should also inform/discuss if other experimental models of cerebral malaria, if any, also course with the observed alterations concerning decrease in body temperature, neutrophilia and increase in NLR. This information could appear as limitations of the study developed.

Thank you for your comment. We have taken your observations into consideration and have included additional text in our manuscript discussion (line 245-255) to address the issues raised.

  1. It might be interesting to perform all the experiments done with the two strains of mice by initiating malaria infection with sporozoites. According to reference 43, humans infected with sporozoites showed a marked increase in NLR as soon as parasites appear in the blood stream.

We agree that investigating the effects of the onset of infection with sporozoites in both mouse strains would be an interesting avenue. Exploring this aspect could provide valuable information on the dynamics of the immune response and the possible role of neutrophils in Plasmodium infection. Unfortunately, we do not have the structure to perform these experiments at the moment, but we will take this into consideration in future studies.

  1. The description of the statistical analyses performed should be improved.

We agree with your comment, and in the revised version of the manuscript we have provided a more detailed and comprehensive description of the statistical methods employed, including the specific tests used, the confidence level, and the adjustment made for multiple comparisons.

The complete letter, containing the responses to all reviewers, and the revised version of our manuscript are attached.

Reviewer 3 Report

This is a correct paper using murine models of experimental cerebral malaria (ECM) and non-cerebral experimental malaria. The authors found that clinical signs of ECM occur suddenly in Plasmodium infected C57BL/6 mice (models for ECM) and are rapidly followed by their death, however, the parasitemia did not differ from that of the mice model of non-cerebral experimental malaria. Moreover, the development of ECM is preceded by an increase in the neutrophil percentage and neutrophil-T cell ratios in the spleen and blood of ECM mice. The conclusions are supported by the experimental data. The discussion is limited to stating that the authors' results are consistent with some previous data, but not with some other ones.

Author Response

Reviewer 3

“This is a correct paper using murine models of experimental cerebral malaria (ECM) and non-cerebral experimental malaria. The authors found that clinical signs of ECM occur suddenly in Plasmodium infected C57BL/6 mice (models for ECM) and are rapidly followed by their death, however, the parasitemia did not differ from that of the mice model of non-cerebral experimental malaria. Moreover, the development of ECM is preceded by an increase in the neutrophil percentage and neutrophil-T cell ratios in the spleen and blood of ECM mice. The conclusions are supported by the experimental data. The discussion is limited to stating that the authors' results are consistent with some previous data, but not with some other ones”.

Thank you for reviewing our article. We appreciate your positive feedback and acknowledgement of the study results.  We have carefully considered your comment about the discussion and try to improve it. Once again, thank you for your valuable comments on our work.

The complete letter, containing the responses to all reviewers, and the revised version of our manuscript are attached.
